# Predicting geographic location from genetic variation with deep neural networks

CJ Battey*, Peter L Ralph, Andrew D Kern

University of Oregon, Institute of Ecology and Evolution, Eugene, United States

**Abstract** Most organisms are more closely related to nearby than distant members of their species, creating spatial autocorrelations in genetic data. This allows us to predict the location of origin of a genetic sample by comparing it to a set of samples of known geographic origin. Here, we describe a deep learning method, which we call Locator, to accomplish this task faster and more accurately than existing approaches. In simulations, Locator infers sample location to within 4.1 generations of dispersal and runs at least an order of magnitude faster than a recent model-based approach. We leverage Locator's computational efficiency to predict locations separately in windows across the genome, which allows us to both quantify uncertainty and describe the mosaic ancestry and patterns of geographic mixing that characterize many populations. Applied to whole-genome sequence data from *Plasmodium* parasites, *Anopheles* mosquitoes, and global human populations, this approach yields median test errors of 16.9km, 5.7km, and 85km, respectively.

**\*For correspondence:**
cjbattey@gmail.com

**Competing interests:** The authors declare that no competing interests exist.

## Introduction

In natural populations, local mate selection and dispersal create correlations between geographic location and genetic variation – each individual's genome is a mosaic of material inherited from recent ancestors that are usually geographically nearby. Given a set of genotyped individuals of known geographic provenance, it is therefore possible to predict the location of new samples from genetic information alone (*Guillot et al., 2016*; *Yang et al., 2012*; *Wasser et al., 2004*; *Rañola et al., 2014*; *Bhaskar et al., 2016*; *Baran et al., 2013*). This task has forensic applications – for example, estimating the location of trafficked elephant ivory as in *Wasser et al., 2004* – and also offers a way to analyze variation in geographic ancestry without assuming the existence of discrete ancestral populations.

The most common approaches to estimating sample locations are based on unsupervised genotype clustering or dimensionality reduction techniques. Genetic data from samples of both known and unknown origin are jointly analyzed, and unknown samples are assigned to the location of known individuals with which they share a genotype cluster or region of PC space (*Breidenbach, 2019*; *Battey et al., 2018*; *Cong et al., 2019*). However, these methods require an additional mapping from genotype clusters or PC space to geography and can produce nonsensical results if unknown samples are hybrids or do not originate from any of the sampled reference populations.

Existing methods for estimating sample location that explicitly model continuous landscapes use a two-step procedure. A smoothed map describing variation in allele frequencies over space is first estimated for each allele based on the genotypes of individuals with known locations, and locations of new samples are then predicted by maximizing the likelihood of observing a given combination of alleles at the predicted location. In methods like SPASIBA (*Guillot et al., 2016*) and SCAT (*Wasser et al., 2004*), allele frequency surfaces are estimated by fitting parameters of a Gaussian function of set form (but see *Rañola et al., 2014* for an alternate approach based on smoothing techniques from image analysis).

Since all such methods use relatedness to other contemporary samples, any information about the location of a new sample necessarily comes from ancestors shared with the reference set. As illustrated in *Figure 1*, we expect a priori, that the genealogical relationships among a set of samples (and therefore the spatial location of ancestors) will vary along the genome. This means that a complete look at geographic ancestry would include not just a point estimate of spatial location, but an estimate of uncertainty that accounts for the partially correlated genealogies of recombining chromosomes.

In the past few years, there has been a explosion in the use of supervised machine learning for population genetics for a number of tasks, including detecting selection (*Schrider and Kern, 2016*; *Mughal and DeGiorgio, 2019*; *Sugden et al., 2018*), inferring admixture (*Schrider et al., 2018*; *Durvasula and Sankararaman, 2019*), and performing demographic model selection (*Pudlo et al., 2016*; *Villanea and Schraiber, 2019*). Applications to population genetics increasingly make use of the latest generation of machine learning tools: deep neural networks (a.k.a. 'deep learning') (*Sheehan and Song, 2016*; *Kern and Schrider, 2018*; *Chan et al., 2018*; *Flagel et al., 2019*;

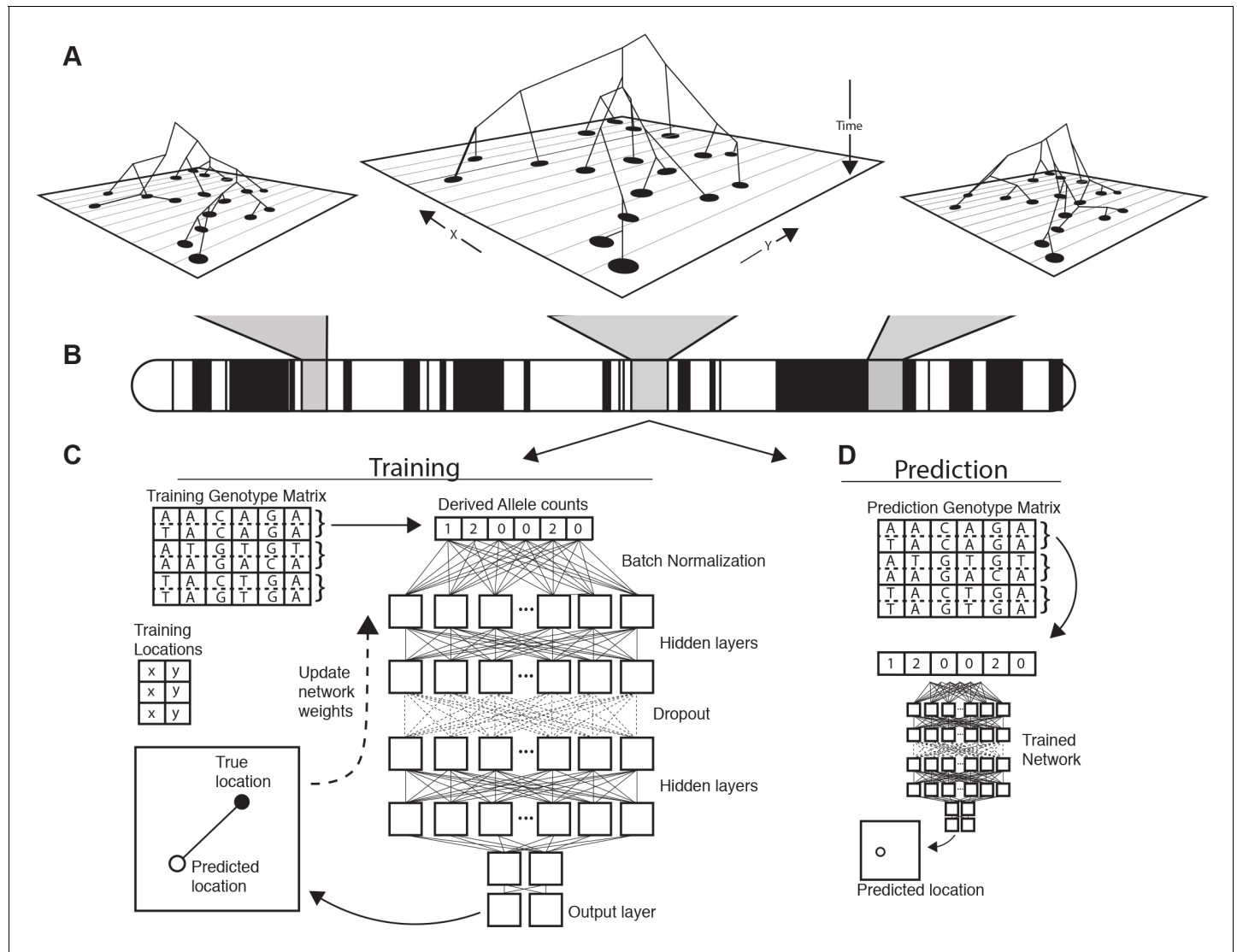

**Figure 1.** Conceptual schematic of our approach. Regions of the genome reflect correlated sets of genealogical relationships (**A**), each of which represents a set of ancestors with varying spatial positions back in time. We extract genotypes from windows across the genome (**B**), and train a deep neural network to approximate the relationship between genotypes and locations using Euclidean distance as the loss function (**C**). We can then use the trained network to predict the location of new genotypes held out from the training routine (**D**).

*Adrion et al., 2020*). A significant feature of neural networks is that they allow the input of raw genotype information, as we perform below, without initial compression into summary statistics.

In this paper, we introduce Locator, a highly efficient deep learning method for the prediction of geographic origin of individuals from unphased genotype data. Locator uses deep neural networks to perform prediction directly from genotypes, but without assuming any explicit model of how genotypes vary over the landscape. Moreover, unlike many modern supervised machine learning methods in population genetics, (e.g. *Kern and Schrider, 2018*) our training set need not be obtained via simulation. We assume only that there is some function relating geographic locations to the probability of observing a given combination of alleles, and use a deep, fully connected neural network to approximate this mapping for a set of genotyped individuals with known locations. The trained network is then evaluated against a set of known individuals held out from the training routine and used to predict the geographic location of new samples based on their genotypes. Applied separately to windows across the genome, Locator also estimates uncertainty in individual-level predictions and can reveal portions of an individual's genome enriched for ancestry from specific geographic areas.

For the empirical population genomic data we analyze here, Locator achieves state-of-the-art accuracy an order of magnitude faster than competing methods. Here, we describe the implementation, test on simulated data, and demonstrate its use in empirical data by estimating sampling locations for *Anopheles* mosquitoes in Africa from the AG1000G project (*The Anopheles gambiae 1000 Genomes Consortium, 2015*), *P. falciparum* parasites from Asia, Africa, and the Americas from the *P. falciparum* community project (*Pearson et al., 2019*), and global human populations from the Human Genome Diversity Project (HGDP; *Bergström et al., 2020*).

## Results

### Locator is fast and accurate

We first evaluated Locator's performance in simulations of populations evolving in continuous space with varying rates of dispersal – an idealized setting in which all alleles should vary smoothly over the map. In *Figure 2* we show that validation error increases along with the dispersal rate of the population. Interestingly, error is roughly constant when measured in terms of the mean per-generation dispersal rate, ranging from 3.16 to 4.09 generations of dispersal given our largest training dataset (450 samples, 100,000 SNPs; *Figure 2—figure supplement 1*; *Supplementary file 1*). This suggests that error primarily reflects the underlying biological processes of dispersal and mate selection rather than simple noise from model fitting.

Increasing the number of training samples or the number of SNPs improves accuracy for all simulations (*Figure 2B*). However, we observed diminishing returns when using over 10,000 SNPs or over 200 training samples. Median error for all simulations was also below 10 generations of dispersal for all but the least-dispersive simulation using just 25 training samples; suggesting that even relatively small training datasets can allow inference of broad-scale spatial locations. We discuss theoretical limits on the accuracy of genetic location estimation in Appendix 1.

We were interested to compare the performance of Locator to that of SPASIBA (*Guillot et al., 2016*), the current state-of-the-art method for geographic prediction of genotype data (*Figure 3*; *Supplementary file 2*). However, we were unable to succesfully run SPASIBA with 100,000 or more SNPs from a simulated dataset or on simulations with dispersal rates of 0.63 or 1.29 map units/generation, due to out-of-memory errors on a 64-bit system with 400 Gb of RAM. We could, however, compare at smaller numbers of SNPs and reduced dispersal. At a mean dispersal distance of 0.45 map units SPASIBA's median test error was slightly lower when run on 1000 SNPs (Wilcoxon test, p=0.009) but results were similar at 100 or 10,000 SNPs. (Wilcoxon test, $p = 0.184$ and 0.936). However, Locator is much faster – training on 10,000 SNPs in less than two minutes while SPASIBA requires around six and a half hours (*Figure 2*). These long run times are caused in part by the large number of training localities in our simulated data, because SPASIBA's run time scales with the product of the number of genetic variants and the number of training localities (*Guillot et al., 2016*).

While the simulations conform well to modeling assumptions of most methods, we can also compare performance on empirical data. By way of example, we applied Locator and SPASIBA to

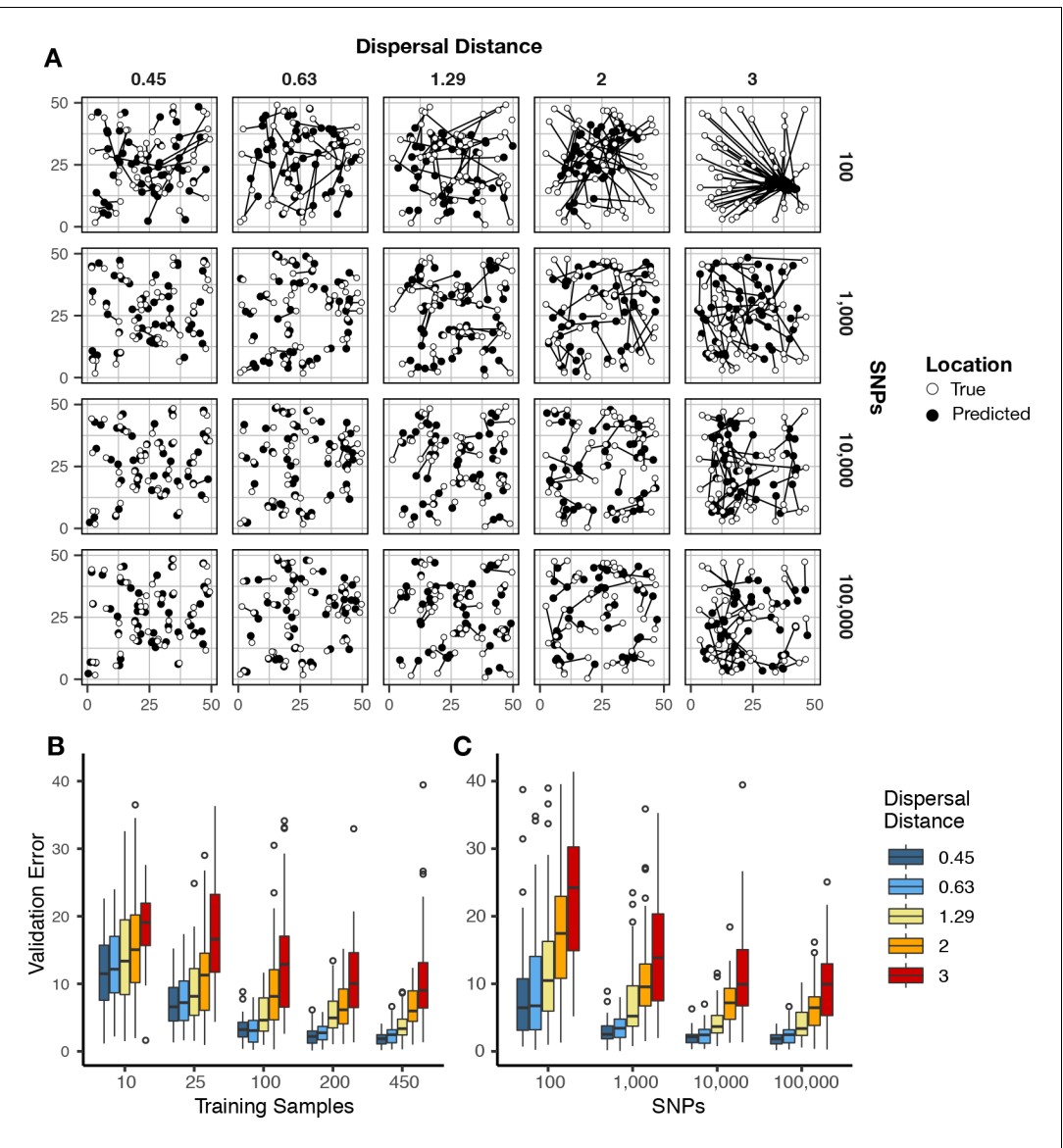

**Figure 2.** Validation error for Locator runs on simulations with varying dispersal rates. Simulations were on a 50 × 50 landscape and error is expressed in map units. (**A**) True and predicted locations by population mean dispersal rate and number of SNPs. 450 randomly-sampled individuals were used for training. (**B**) Error for runs with 100,000 SNPs and varying numbers of training samples. (**C**) Error for runs with 450 training samples and varying number of SNPs. Plots with error in terms of generations of expected dispersal are shown in *Figure 2—figure supplement 1*. The online version of this article includes the following figure supplement(s) for figure 2:

**Figure supplement 1.** Validation error for Locator runs on simulations with varying dispersal distance, expressed in generations of mean dispersal (test error divided by mean dispersal distance per generation).

**Figure supplement 2.** Example training and validation loss histories.

subsets of SNPs from the first five million base pairs of chromosome 2L from the Ag1000G dataset (***Miles and Harding, 2017***; ***Figure 3***). Locator achieves much lower mean error on all runs with more than 100 SNPs, and runs from 3.1x to 532x faster, depending on the number of SNPs. Two factors likely explain this improved accuracy: Locator can handle abrupt changes in allele frequencies across the landscape better than `SPASIBA's` geostatistical model, and the limited number of sampling localities in the AG1000G dataset may act as a prior encouraging Locator to assign samples to specific sampling sites. Maps of predictions from both methods are shown in ***Figure 3—figure supplement 1***. Extrapolating from these run times, running a windowed whole-genome analysis of

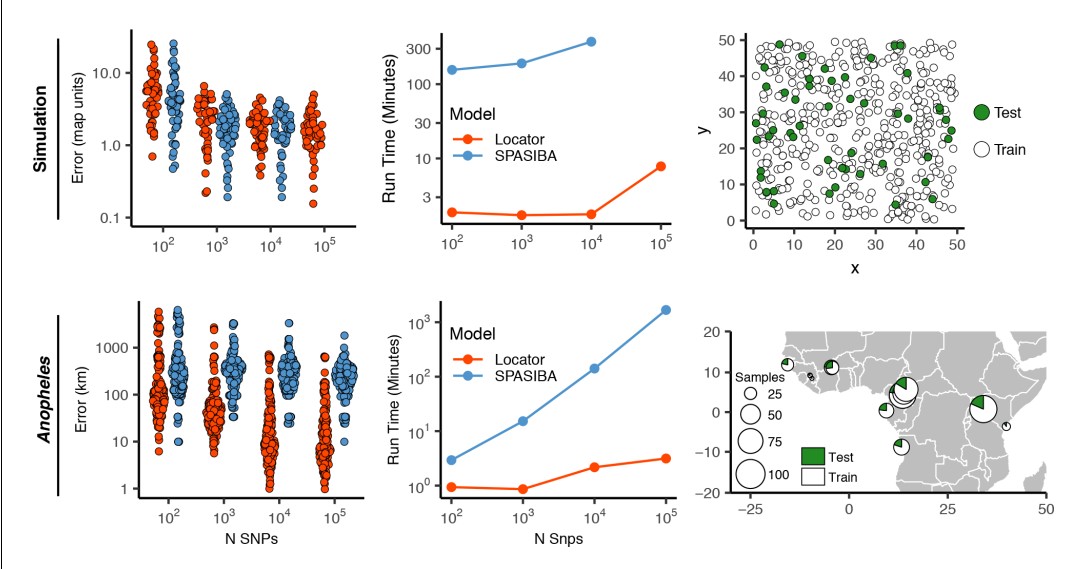

**Figure 3.** Test error and run times for `Locator` and `SPASIBA` on simulated data with dispersal distance equal to 0.45 map units/generation (top; 450 randomly sampled training samples) and empirical data from the ag1000g phase one dataset (bottom; 612 training samples from 14 sampling localities).

The online version of this article includes the following figure supplement(s) for figure 3:

**Figure supplement 1.** Predicted (colored points) and true (black circles) locations for Locator and SPASIBA on the ag1000g dataset.

*Anopheles* in `SPASIBA` would require roughly 70 days of computation on an 80-CPU system for model training alone, versus 3.2 hr on one `GPU` for `Locator`.

## Uncertainty and variation along the genome

By running Locator in windows across the genome we aim to integrate over error associated with the model training procedure while also representing the inherent uncertainty caused by spatial drift of ancestral lineages backwards in time (*Kelleher et al., 2016*). This produces a cloud of predicted locations distributed around the true sample location (*Figure 4*). For individuals near the center of the landscape, these clouds are roughly symmetrical, as expected from our model. Predictions for individuals close to the edge of the landscape appear slightly asymmetrical and are bounded by the true landscape edges, suggesting that our networks have learned the rough shape of the sampled range. The true location was within the 50% contour of a 2d-kernel density surface estimated from the set of per-window predictions for all test samples, demonstrating that this distribution is indeed centered on the true location. We also tested the alternate approach of bootstrapping over a single set of SNPs, which could be useful for smaller datasets or those lacking a reference alignment. Results for this method are discussed in *Figure 4—figure supplement 1*.

Windowed analyses for the three empirical systems we studied are shown in the bottom panels of *Figures 5*, *6*, *7*. We discuss the implications of these predictions for each species in the following sections, but in general we find that the windowed analysis provides a good depiction of uncertainty in a sample's location – either surrounding a single location for samples with low error, or distributed across a wide region including multiple training localities for samples with high error. In several cases, predicted locations also project in the direction of known historic migrations (as in human data), or are split among localities shown in previous analyses to experience high gene flow (as in *Anopheles*).

We summarize genome-wide window predictions in two ways: 1) by taking a kernel density estimate of the predictions and then finding the point of maximum density, and 2) by computing the centroid of the windowed predictions. These estimates are similar in spirit to ensemble prediction methods (*Ho and forests, 1995*; *Breiman, 1996*), but should not be interpreted as true statistical confidence intervals. At least in the context of windowed analyses, differences in predictions among windows appears to primarily reflect variation in ancestry rather than uncertainty in the inference

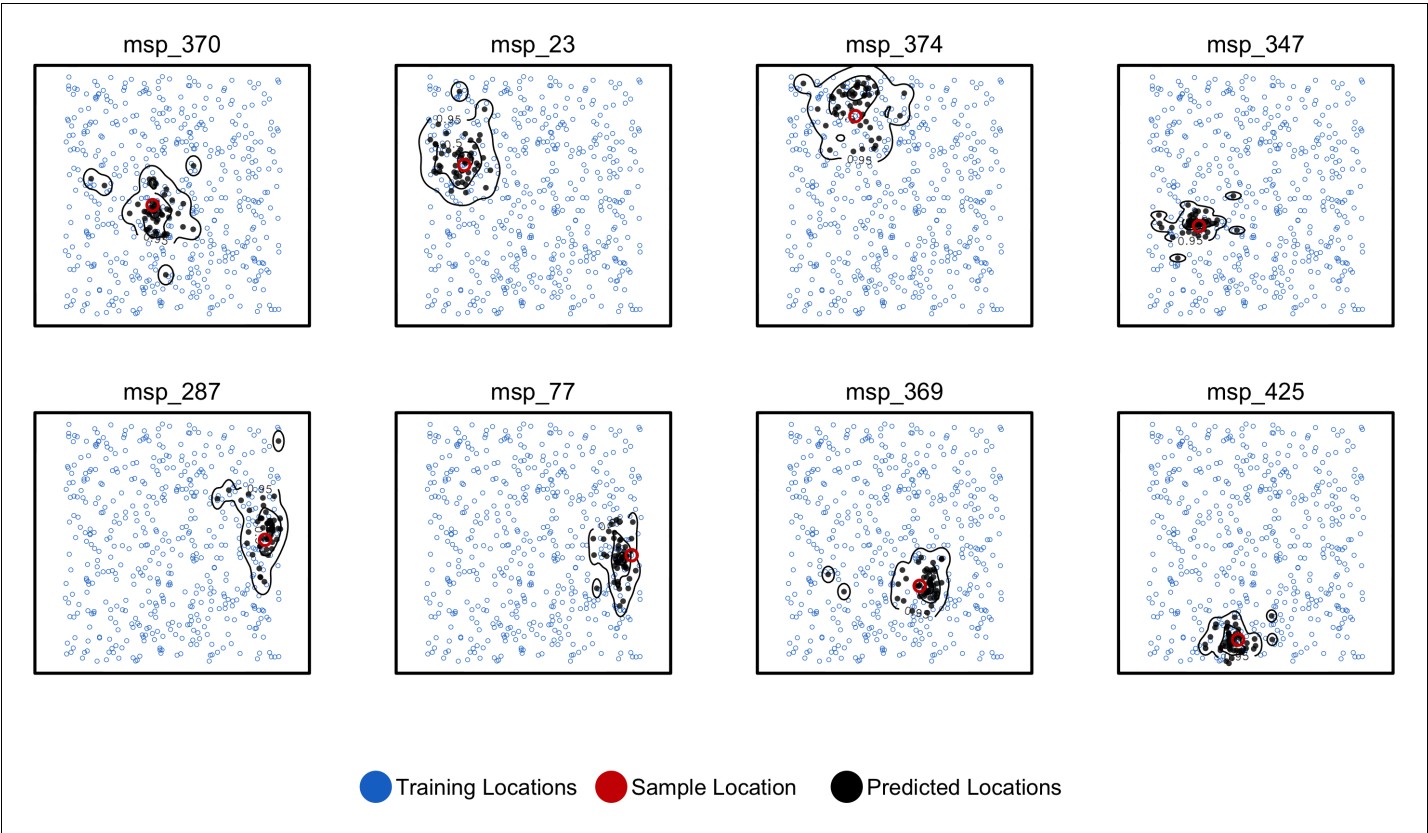

**Figure 4.** Predicted and true locations for eight individuals simulated in a population with mean per-generation dispersal 0.45 (roughly 1% of the landscape width). Black points are predictions from 2Mbp windows, blue points are training sample locations, and the red point is the true location for each individual. Contours show the 95%, 50%, and 10% quantiles of a two-dimensional kernel density across all windows.

The online version of this article includes the following figure supplement(s) for figure 4:

**Figure supplement 1.** Predicted and true locations for eight individuals simulated in a population with an expected dispersal rate of 0.63 map units/ generation, using a set of 10,000 randomly sampled SNPs.

itself, so we suggest the intervals returned by Locator's kernel density estimation are best interpreted as representing areas from which a given proportion of the genome is likely to have originated. In general, we found that the maximum kernel density estimator has lower error, but tends to show classification behavior more than the centroid estimator – snapping to a single training locality rather than interpolating between sets of localities for samples with variable window predictions.

## Empirical analysis

### *Anopheles* mosquitoes

We next turn our attention to the application of Locator to empirical population genomic datasets. In *Figure 5*, we show predicted and true locations for 153 individuals from the Ag1000g dataset of *Anopheles gambiae* and *A. coluzzii*, estimated in 2Mbp windows across the genome. The location with highest kernel density across all windows had a median error of 5.7 km, and the centroid of the per-window predictions had a median error of 36 km (*Supplementary file 3*). Significant prediction error occurs only between sites in Cameroon, Burkina Faso, and the Republic of Guinea – localities which were also assigned to a single ancestry cluster in the ADMIXTURE analysis in *Miles and Harding, 2017*. However uncertainty for these samples was relatively well described by visualizing the spread of per-window predictions, with predicted locations generally lying between sets of localities. The true locality was within the 95% interval of the kernel density across all windows for all samples.

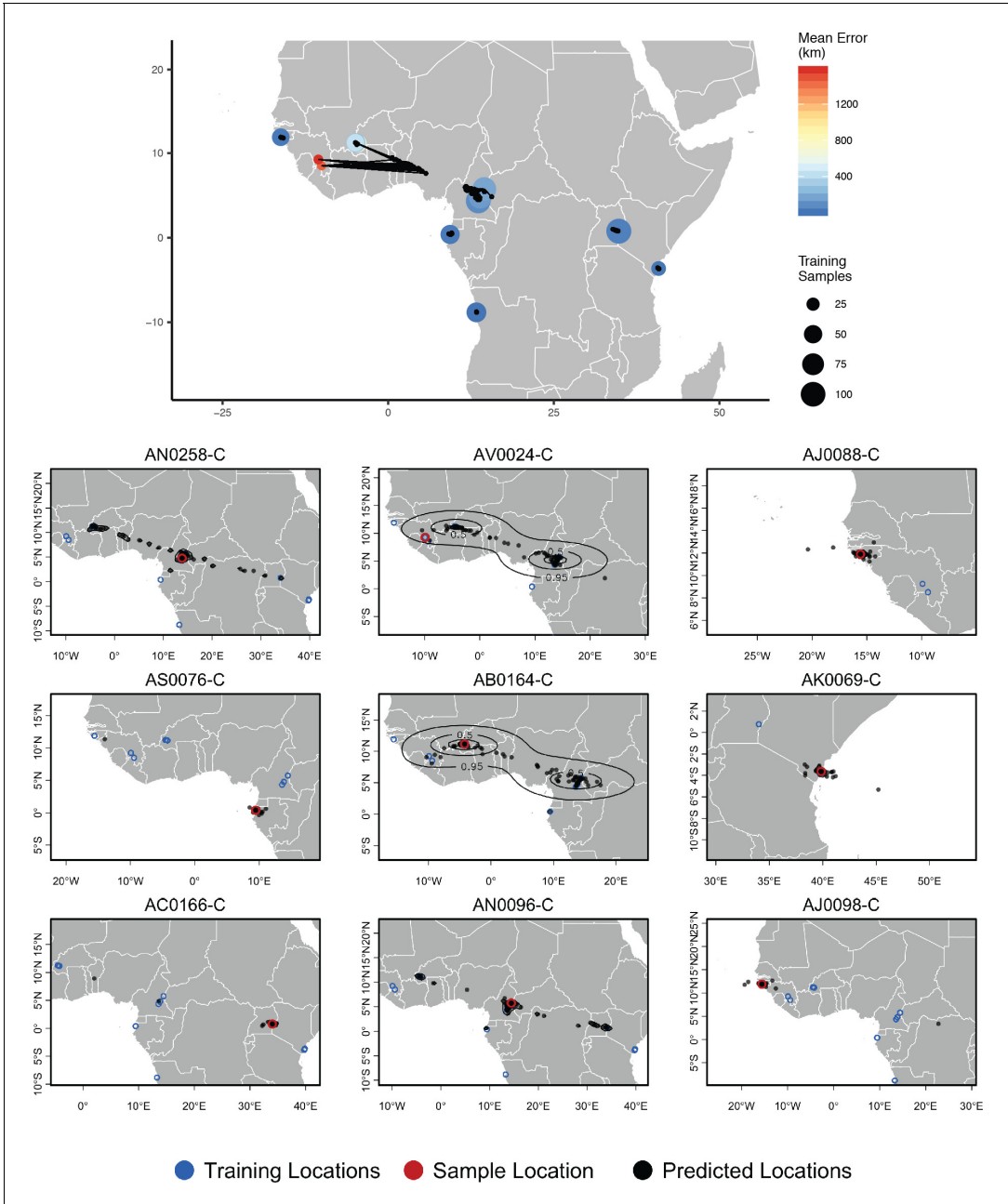

**Figure 5.** Top – Predicted locations for 153 *Anopheles gambiae/coluzzii* genomes from the AG1000G panel, using 612 training samples and a 2Mbp window size. The geographic centroid of per-window predictions for each individual is shown in black points, and lines connect predicted to true locations. Sample localities are colored by the mean test error with size scaled to the number of training samples. Bottom – Uncertainty from predictions in 2Mbp windows. Contours show the 95%, 50%, and 10% quantiles of a two-dimensional kernel density across windows.

The online version of this article includes the following figure supplement(s) for figure 5:

**Figure supplement 1.** Comparison of cross-validation performance on the ag1000g dataset using SNPs from chromosome 3R, under varying network architectures and numbers of SNPs.

**Figure supplement 2.** Performance on 10,000 SNPs from chromosome 2L in the ag1000g phase one dataset when all samples from localities in the true country are dropped from the training set.

**Figure supplement 3.** Performance on 10,000 SNPs from chromosome 2L in the ag1000g phase one dataset when all samples from the true locality are dropped from the training set.

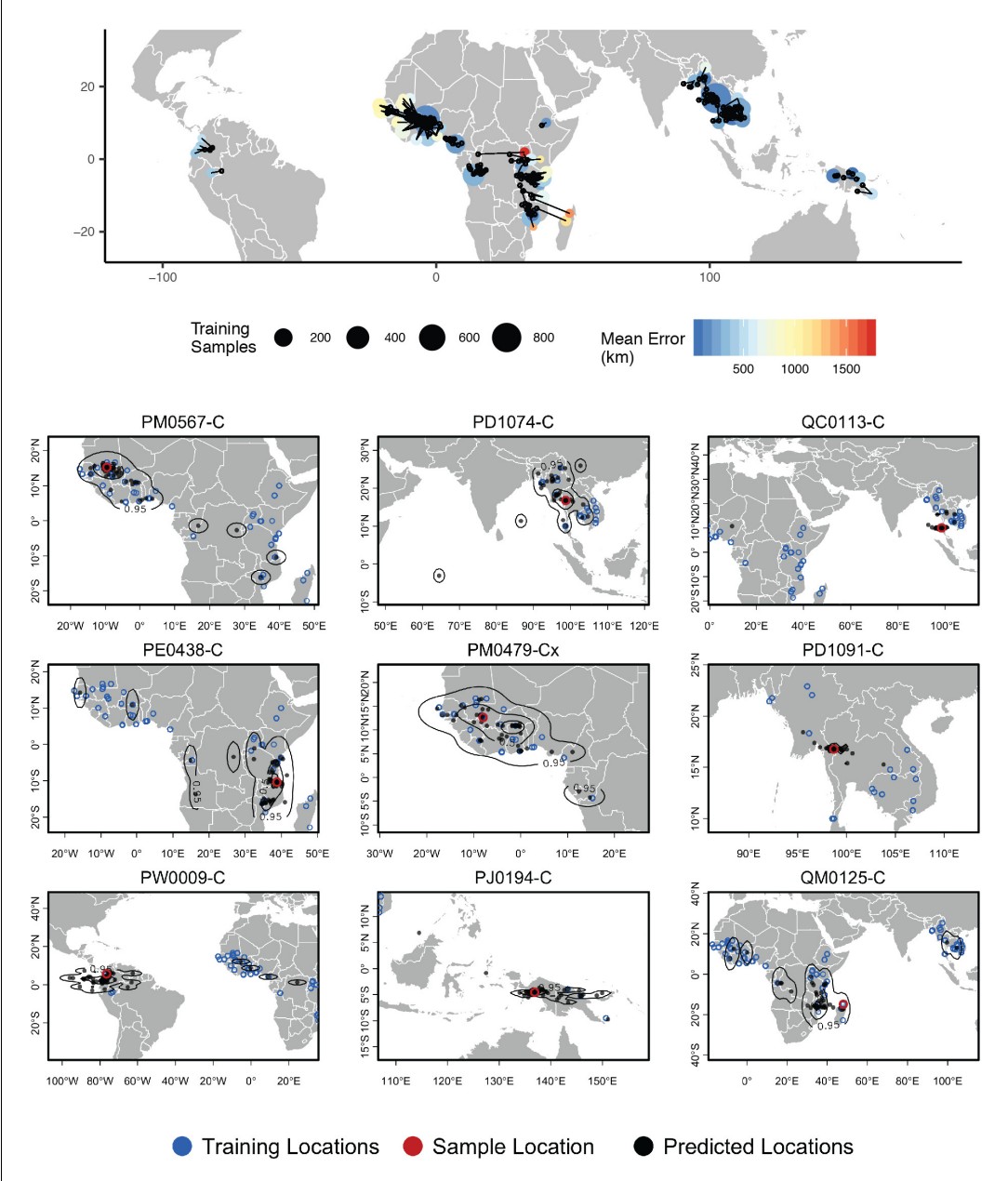

**Figure 6.** Top – Predicted locations for 881 *Plasmodium falciparum* from the *Plasmodium falciparum* Community Project (*Pearson et al., 2019*) (5% of samples for each collecting locality), using 5084 training samples and a 500Kbp window size. The geographic centroid of per-window predictions for each individual is shown in black points, and lines connect predicted to true locations. Sample localities are colored by the mean test error with size scaled to the number of training samples. Bottom – Uncertainty from predictions in 500Kbp windows. Contours show the 95%, 50%, and 10% quantiles of a two-dimensional kernel density across windows.

The online version of this article includes the following figure supplement(s) for figure 6:

**Figure supplement 1.** Centroid prediction error as a function of within-host diversity ($F_{WS}$) for the *Plasmodium falciparum* dataset.

## *Plasmodium falciparum*

In a windowed analysis of *P. falciparum*, Locator's median error is 16.92 km using the maximum kernel density and 218.99 km using the geographic centroid of window predictions (*Figure 6*; *Supplementary file 3*). Mean predicted locations across all windows consistently separate populations in the Americas, West Africa, East Africa, southeast Asia, and Papua New Guinea; consistent with the major population subdivisions described via PCA in *Pearson et al., 2019*. We also see

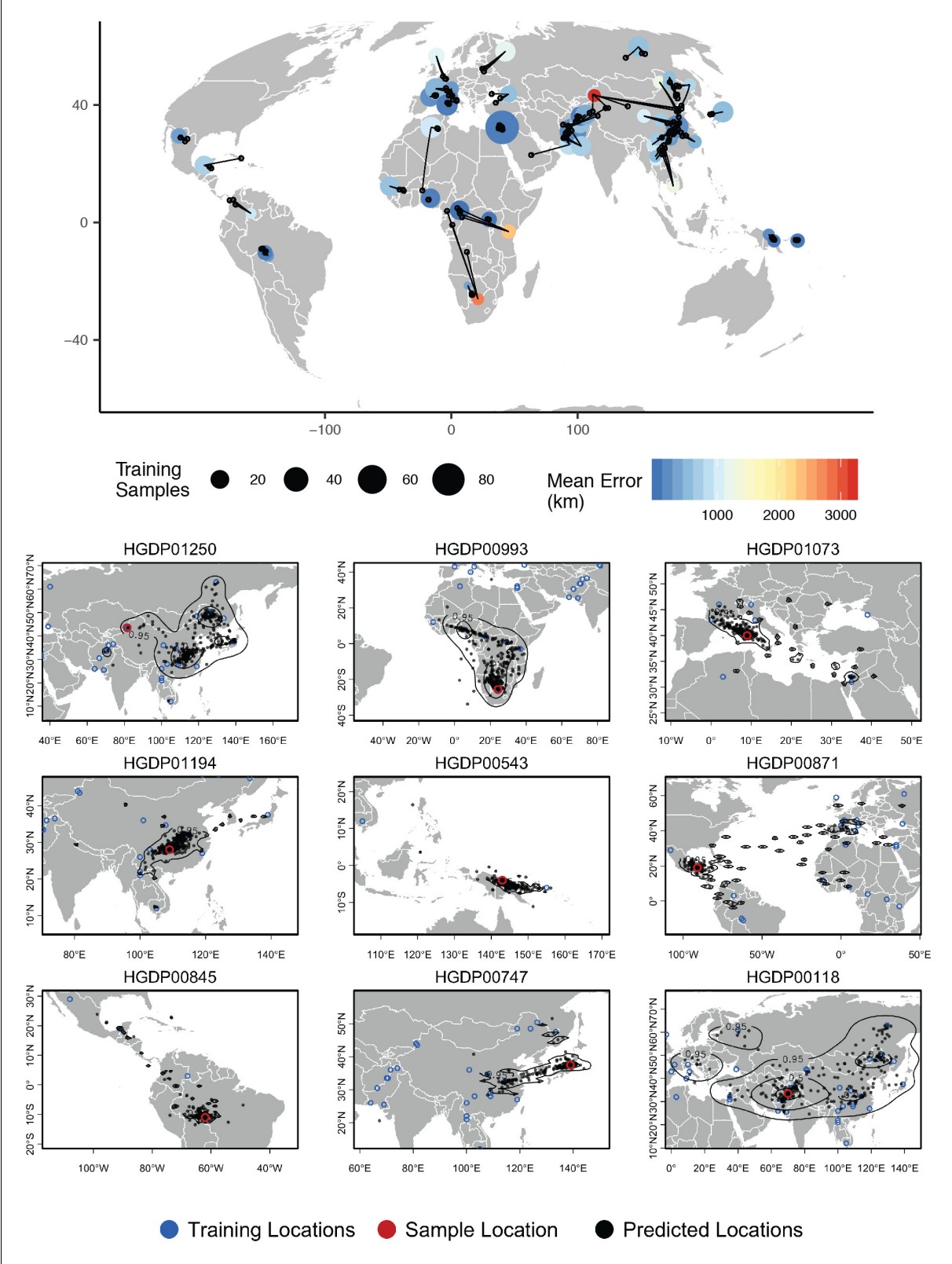

**Figure 7.** Top – Predicted locations for 162 individuals from the HGDP panel, using 773 training samples and a 10Mbp window size. The geographic centroid of per-window predictions for each individual is shown in black points, and lines connect predicted to true locations. Sample localities are colored by the mean test error with size scaled to the number of training samples. Bottom – Uncertainty from predictions in 10Mbp windows. Contours show the 95%, 50%, and 10% quantiles of a two-dimensional kernel density across windows.

*Figure 7 continued on next page*

*Figure 7 continued*

The online version of this article includes the following figure supplement(s) for figure 7:

**Figure supplement 1.** Outliers in windowed Locator analyses identify genomic regions enriched for admixed ancestry.

good discrimination within clusters, particularly in southeast Asia where the average test error is less than 200 km for all but two localities. Error is highest in West Africa, where mean predictions tend towards the center of a set of regional collecting localities (*Figure 6*). These patterns are consistent with previous findings of fine-scale spatial structure in *P. falciparum* in Cambodia (*Miotto et al., 2013*) and low levels of relative genetic differentiation (as measured by $F_{ST}$) in Africa (*Pearson et al., 2019*).

Rates of mixed-strain infection are elevated in West Africa relative to Southeast Asia (*Zhu et al., 2019*; *Pearson et al., 2019*), which we hypothesized could explain the higher prediction error in this region. To test this effect, we plotted Locator's centroid prediction error as a function of within-host diversity ($F_{WS}$; *Auburn et al., 2012*). $F_{WS}$ measures the proportion of population genetic diversity present in individual hosts, with a value of 0 representing maximum within-host diversity and one minimum within-host diversity. If mixed-strain infections explain outliers of prediction error, we would expect that samples with the highest prediction error had low $F_{WS}$. Instead we found a weak positive relationship (*Figure 6—figure supplement 1*), with the highest prediction errors seen in samples with maximum $F_{WS}$ (i.e. minimum infection diversity). Test error then likely reflects low levels of differentiation within *Plasmodium* lineages in West Africa rather than local prevalence of mixed-strain infections.

Again we found that visualizing per-window predictions reflects expected patterns of uncertainty in samples with high mean prediction error. For example, sample QM0215-C was collected in Madagascar and has a mean predicted location in Mozambique, but the spread of per-window predictions indicates a 95% interval that includes the true locality (*Figure 6*, bottom right).

The good performance we observed on this dataset also highlights a strength of `Locator`'s model-free approach. Recall that the sequencing strategy of preparing libraries from human blood samples suggests variant calls represent binned allele frequencies across the population of *Plasmodium* in a human blood sample rather than SNPs in a single *Plasmodium* individual. From the perspective of the network, however, the input genotypes are simply a set of normalized vectors, and the network can approximate the relationship between these vectors and the spatial location of training samples regardless of the generative process.

## Human populations

For humans in the HGDP dataset, the location with highest kernel density across all windows has a median test error of 85 km, and the centroid of window predictions has a median error of 452.6 km (*Figure 7*, *Supplementary file 3*). Visualizing the geographic distribution of predictions across the genome shows that predictions tend to cluster around the true reported sampling location, but also extend toward other sampling locations in a manner that reflects known patterns of human migration.

For example, the two largest individual errors in our analysis are found in South African Bantu individuals and Xibo people from western China. Predicted locations of South African Bantu people project towards the historic source of Bantu migrations in west Africa (*de Filippo et al., 2012*), with some regions of the genome also projecting in the direction of east African Bantu populations (*Figure 7*, sample HGDP00993). In the case of Xibo people from western China Locator consistently predicts locations in Manchuria, central China, and southern Siberia – significantly east of the true sample location. This may reflect the known movement of this population, which historically originated in Manchuria and was resettled in western China during the 18th century (*Gorelova, 2002*; *Zikmundová, 2013*; *Figure 7*, sample HGDP01250). A sample of individual-level predictions is included in *Figure 7*.

To test whether outlier geographic predictions reflect error in the model fitting procedure versus true variation in ancestry in a given region of the genome, we ran principal component analyses on windows for which a Maya individual (sample HGDP00871) has predicted locations in Europe and Africa. In these windows, the Maya sample clusters with other individuals from the regions predicted

by Locator – western Europe and Africa, respectively – rather than with other individuals from the Americas (*Figure 7—figure supplement 1*). This suggests outlier predictions primarily reflect variation in ancestry in different regions of the genome, rather than stochastic error in model fitting.

## Genomic factors affecting prediction accuracy

Locator's prediction accuracy varied widely along the human genome. To assess the sources of this variation, we first examined how recombination rate interacts with the accuracy of `Locator` predictions generated from different regions of the genome. A priori we might expect recombination rate to affect accuracy because in regions of the genome with higher recombination, there are a greater number of distinct genealogies, and hence a given sample has inherited from a larger subset of the possible ancestors. Test error was estimated as the distance in kilometers from the true sampling location to the geographic centroid of the cloud of per-window predictions, and is shown in *Figure 8* plotted against local recombination rates from the HapMap genetic map (*International HapMap Consortium, 2003*). We find a relatively strong negative correlation ($p<0.0001$, $R^2 = 0.27$) – windows with the lowest recombination rates in general have the highest prediction error. This is consistent with our expectation that regions of the genome representing a greater number of marginal genealogies will yield more accurate predictions of a sample's location.

This finding suggests recombination-based distances may be a better basis for establishing window bounds. We tested this by replicating our analysis of `HGDP` samples using a `10-centiMorgan` window size, which yields approximately the same number of windows across the genome. We found the distribution of centroid prediction errors across windowing methods was very similar (*Figure 8—figure supplement 3*) but the variance in per-window prediction error was lower when using recombination-based windows (*Figure 8—figure supplement 4*; Levene's test $p = 0.0002$, df = 654).

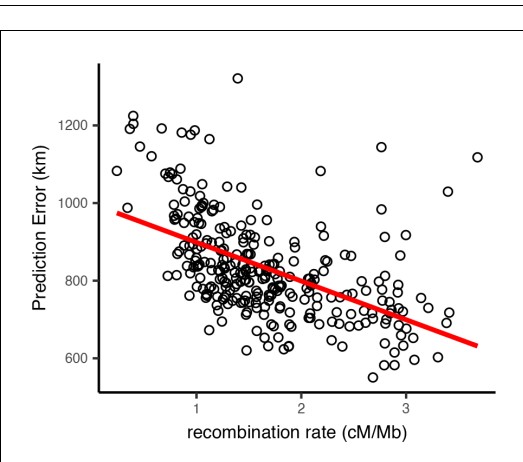

**Figure 8.** Per-window test error and mean recombination rate for human populations in the HGDP dataset. The top 2% of windows by test error were excluded from this analysis. The slope of the least-squares linear fit is −99.9723 km/(cM/Mbp) and has adjusted $R^2 = 0.2704$.

The online version of this article includes the following figure supplement(s) for figure 8:

**Figure supplement 1.** Mean test error for HGDP samples in 10-megabase windows.

**Figure supplement 2.** Mean test error for HGDP samples in 10-centimorgan windows.

**Figure supplement 3.** Distributions of centroid prediction error across samples.

**Figure supplement 4.** Distributions of prediction error across windows when using megabase- versus centimorgan-based windows.

Examining plots of window error along the genome suggests much of this effect is caused by high error in windows containing a centromere when using physical distance-based windows (*Figure 8—figure supplement 1*, *Figure 8—figure supplement 2*). Centromere presence explains 26.9% of variance in window prediction error using 10 Mb windows, but just 2.6% of variance with centiMorgan-based windows. Thus, we recommend researchers interested in analyzing individual window predictions use recombination-based windows when possible, but find that using fixed windows is sufficient for estimating individual-level locations.

Last, to examine how predictions vary near a well-known geographically differentiated region of the genome, we trained a `Locator` model using SNPs from within 100 kb of the *EDAR* gene. The rs3827760 SNP is a $A \rightarrow G$ mutation that created a derived allele which has reached high frequency in East Asian and North American populations but is rare elsewhere (*Bryk et al., 2008*). This variant is thought to be associated with traits including hair thickness (*Fujimoto et al., 2008*), and the EDAR region has been proposed as a site of recent positive selection in several analyses (*Voight and Kudaravalli, 2006*; *Tang et al., 2007*; *Williamson et al., 2007*). We focused on predictions for samples from central Asia, in the middle of the east-west cline in rs3827760 allele frequencies across Eurasia.

In this analysis, the direction of prediction error generally followed the genotype at rs3827760 – individuals with a G allele tend to predict east of their true location (*Figure 9*). This was particularly clear in heterozygous populations. For example, homozygous G/G Xibo and Hazara individuals predict east of their true location, while one homozygous A/A Uyghur sample predicts well west of the sampling site. 3 of 4 heterozygous samples also predict east of their true location, and five other A/A Central/South Asian samples predict significantly west. The discrepancies between sampling and predicted locations likely in fact represent signal: haplotypes carrying a G allele at this locus likely have more 'close' relatives in eastern than western Eurasia. These patterns are magnified versions of the trends seen in windowed analyses, suggesting that strong differentiation in this genomic region biases location predictions away from the center of the geographic cline.

## Effects of unsampled populations

To understand how Locator predictions vary when a sample's true locality is not included in the training set, we ran analyses on a single window of the *Anopheles* data at two scales – first dropping only sites from a specific sampling location, and then dropping all sites from a given country (*Figure 5—figure supplements 2–3*). Prediction error is much higher for individuals from regions excluded from training – increasing from a median of 14 km when training and test samples are randomly split to 116 km when excluding individual localities, and 778 km when excluding whole countries.

In most cases, predicted locations appear to project toward the nearest locality included in the training set (*Figure 5—figure supplement 3*). This is particularly the case when populations at the edge of the map are excluded. Locator networks appear to learn something about the boundaries of the landscape based on the distribution of training points, and show a tendency to project towards the middle of the landscape when given a small number of SNPs (e.g. the top right panel of *Figure 2A*), a trivial optimization of the loss function. We also see evidence of Locator learning

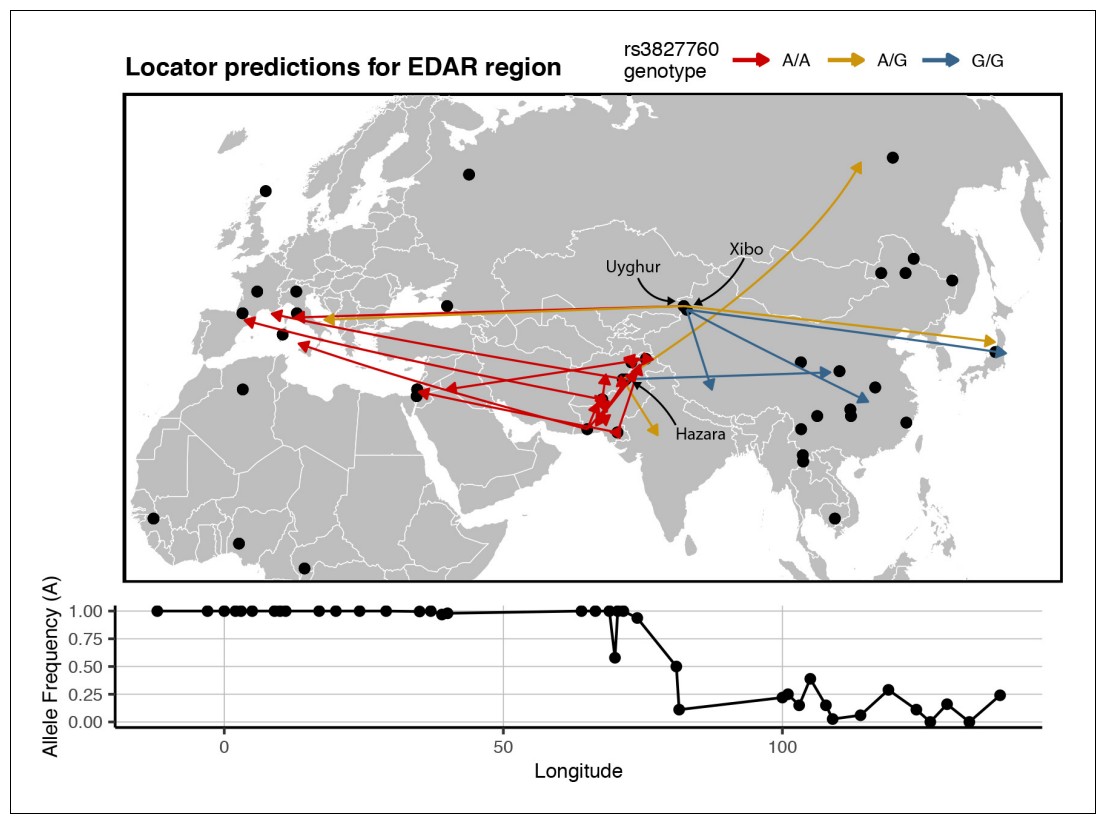

**Figure 9.** Predicted locations for HGDP samples from central Asia using a model trained on SNPs within 100 kb of *EDAR*. Black points show sampling locations. Arrows are colored by genotype at variant rs3827760 and point towards the predicted location. Frequency of the A allele by longitude is shown below the map.

some nonlinear aspects of population structure in the sample. For example, when Angolan *A. coluzzii* are excluded from the training set many of their predicted locations project toward the *A. coluzzii* sample localities in Burkina Faso rather than the much closer sampling localities for *A. gambiae* in Cameroon and Gabon. In general, we find that Locator can interpolate unsampled localities relatively well when genetic differentiation is smooth over the landscape (as among *A. gambiae* localities in west Africa), but does not extrapolate outside the bounds of the training set. Sampling the full landscape, or at least a sufficient portion thereof, is thus an important consideration in running our method.

## Discussion

The correlation of genealogy and geography leaves genetic signals of ancestral location across the genome that one can leverage for practical inference. For instance, tracking the migratory routes of disease vectors such as *Anopheles* (*Huestis et al., 2019*) could in principle be achieved if one could accurately predict origin from DNA sequence data. Similarly, establishing the location of origin from biological samples is critical to anti-poaching conservation efforts (*Wasser et al., 2004*), and efforts to map transmission during the ongoing SARS-CoV-2 pandemic have been informed by analysis of geographically restricted genetic variants (see e.g., https://nextstrain.org/ncov/global). In this report, we present a new tool, Locator, which uses a deep neural network to predict the geographic location of a sample on the basis of its genotype. We show that Locator is highly accurate, computationally efficient, and can scale to thousands of genomes.

In simulations we found that our method achieves similar accuracy as a state-of-the-art model-based approach, SPASIBA (*Guillot et al., 2016*), and does so at least an order of magnitude faster. We show that the accuracy of our estimator is naturally measured in terms of the dispersal rate of the population and that predictions from Locator are consistently within 3–4 generations of mean dispersal across a wide range of dispersal distances (*Figure 2*, *Figure 2—figure supplement 1*). However, the greatest increase in accuracy relative to SPASIBA was in empirical data (*Figure 3*). This seems to reflect two aspects of our approach that highlight its strengths and weaknesses: as a process-agnostic model Locator can easily handle situations in which allele frequencies do not vary smoothly over the landscape. However, the spatial concentration of the *Anopheles* samples may act as a strong prior that incentivizes the network to predict sample locations near sampling localities (*Figure 3*, *Figure 5*), thus acting more as a classifier than a continuous predictor. Thus, sampling should be taken into account when interpreting Locator's output, and when possible users should avoid highly clustered sampling regimes.

Locator's computational efficiency makes it practical to estimate uncertainty through resampling approaches like windowed analysis or bootstrapping over the complete genotype matrix. The full windowed analysis of the HGDP data took roughly 30 hr to run on a single GPU, and windowed analysis of all 5965 complete *Plasmodium* genomes took just 8 hr. Thus, training Locator models for biobank-scale datasets including whole genomes of tens or hundreds of thousands of samples is well within reach, particularly if windows can be run on separate GPUs. This allows us to estimate uncertainty in predicted locations due both to our prediction methodology as well as biology; with repeated training runs integrating over error associated with network training and prediction and the windowed analysis allowing us to predict geographic origins for regions of the genome reflecting distinct sets of genealogical relationships.

Disentangling these sources of error is challenging, but analysis of human data for which we have strong prior knowledge of recent population movements suggests that much of the variation in genome-wide prediction we see reflects historic patterns of migration rather than simple prediction error. For example, genomes from Hazara individuals in central Asia return predicted locations extending from central Asia to Mongolia (*Figure 7* bottom, sample HGDP00118), which is consistent with historic records (*Qamar et al., 2002*), previous analysis of Y chromosome data (*Zerjal et al., 2003*), and identity-by-descent tract sharing (*Lawson et al., 2012*) all of which find evidence of recent shared ancestry between Mongolian and Hazara individuals. Similarly some Maya individuals found to have a small proportion of European ancestry in previous analyses *Rosenberg et al., 2002* have predicted locations extending from central Mexico across the Atlantic to Europe and west Africa in windowed Locator analysis (*Figure 7* bottom, sample HGDP00871), and these signals are replicated in principal components analysis (*Figure 7—figure supplement 1*).

The correspondence between our explicitly geographic method and unsupervised clustering or dimensionality reduction methods highlights the implicit prior assumption of genetic-geographic correlation often made when interpreting the output of `STRUCTURE` or `PCA`. Rather than mapping population structure to geography as a post-hoc interpretation, Locator and other continuous assignment methods directly incorporate space in the model. This also points to a critical consideration in running any form of supervised population clustering. Information about population structure comes only from the relative relationships among training and test samples, and interpretations can only be made relative to the set of training samples used. In the case of the HGDP panel, samples were intentionally selected to cover what were thought to be distinctive populations reflecting a vaguely pre-modern distribution of human genetic diversity (*Harry and Marks, 1999*), and so would probably not be a good reference set for random individuals drawn from areas with recent histories of large population movements such as the United States.

Here, we have shown that our method, `Locator`, is fast, accurate, and scales well to large samples. However, we see several next steps that could improve the approach. First, our current implementation uses only diploid genotypes and does not pass the network any direct information about haplotype structure. Incorporating SNP position information and phased haploid sequences would likely increase inferential power, as in the case of unsupervised clustering (*Lawson et al., 2012*). Running this method on phased, haploid genomes could also in theory allow us to predict parental locations individually for loci with ancestry from different geographic regions. Second, our network currently uses a simple fully connected architecture; it could be that other network architectures such as recurrent neural networks might be better suited for this task (e.g. *Adrion et al., 2020*). Indeed the application of deep learning to population genetics is still in its infancy and we imagine much progress will be made in the coming years along these lines.

A central issue in this task will be understanding the limits of process-agnostic inference: when and how can we best use machine learning approaches in population genetics? `Locator` is essentially a flexible regressor designed without direct reference to genealogical or genetic process, so the exact design of the network or the arrangement of the weights and biases in a trained model gives little insight into the underlying mechanisms driving its inferences. These are instead best assessed by carefully studying the natural history of the focal organism, and by interpreting model output with population genetic theory and simulations. We suggest machine learning methods be seen as tools to answer narrowly defined questions, used as a complement to process-based statistical models that give deeper insight into the mechanisms generating the data being analyzed.

In principle, a completely process-based inference method that accurately modeled spatial population dynamics might provide the most accurate and best-calibrated inference of location and/or ancestry. However, the problem of modeling spatial populations is not solved, and all existing methods are based on assumptions (e.g. an unchanged species range on a uniform landscape [*Wright, 1943*]) that may not hold in practice. For this reason, machine learning approaches might outperform methods based on process. Beyond the present application we believe a powerful approach will be to combine process-based population genetic models with the generalization abilities of deep learning – analogous to natural language processing methods that supplement neural network word embeddings with syntax models (*Strubell and McCallum, 2018*). This could allow us to leverage theory where assumptions are well-founded while turning to robust generalized optimization techniques when they are not.

## Materials and methods

### Preprocessing

`Locator` transforms input data in VCF or Zarr format to vectors of allele counts per individual using the scikit-allel (*Miles and Harding, 2017*) and numpy (*van der Walt et al., 2011*) libraries. Sites with missing data are replaced with two draws from a binomial distribution with probability equal to the frequency of the derived allele across all individuals – a discrete version of the common practice of assigning missing data as the mean allele frequency in genotype PCAs (e.g. the default settings for PCA in the R package adegenet [*Jombart, 2008*]). We provide functions for filtering SNPs based on minor allele count, and by default remove singleton sites from the alignment prior to model fitting. The geographical $x$ and $y$ coordinates are scaled to have mean 0 and variance one prior to training,

while allele counts are scaled prior to model fitting by a batch normalization layer within the network. Batch normalization Z-normalizes activations of a neural network during training to reduce shifts in the distribution of parameter values across batches, which allows faster learning rates and sometimes reduces overfitting (*Ioffe and Szegedy, 2015*).

`Locator` selects user-defined fraction of the samples with known locations to use in training the model (the default is 0.9); remaining samples with known locations are kept aside as 'validation' samples. The validation set is used to tune the learning rate of the optimizer and set the stopping time of model training, but does not directly contribute to the loss used to fit model parameters. Throughout this manuscript, we use 'validation loss' to refer to error estimated on the validation set, and 'test error' to refer to error calculated on a set of samples entirely held out from the model training procedure.

For datasets with small sample sizes the random train/test split may lead to some regions being under- or overrepresented in the training sample. To mitigate this we suggest fitting multiple models with different random seeds, yielding an ensemble of models trained on different subsets of the original dataset. Predictions from this ensemble can then be summarized in the same way as windows or bootstrap samples (see below). An example of this approach is included in the Locator documentation (https://github.com/kern-lab/locator).

## Network

We use the unphased, diploid genotype vector of each individual as input to the network, whose target output is the two-dimensional coordinates of that individual in space. Locator uses a deep neural network consisting of a stack of fully connected 'dense' layers, implemented using the Keras (*Chollet, 2015*) frontend to tensorflow (*Abadi et al., 2015*). Roughly speaking, the network is trained to estimate a nonlinear function mapping genotypes to locations using gradient-based optimization. Models start with randomized initial parameters and are fit to data by looping through the training set and iteratively adjusting the weights and biases of the network. We use an early stopping function to monitor loss during training and under default settings stop training runs when validation loss has not improved for 100 epochs. We also use a learning rate scheduler to decrease the learning rate of the optimizer when validation loss stops improving, which we found to be effective in preventing the trajectories of training and validation loss from diverging. The program also outputs a plot of training and validation loss after each training run (*Figure 2—figure supplement 2*).

Locator's architecture uses a batch normalization layer followed by a sequence of fully connected layers with a dropout layer in the middle of the network (*Figure 1*). The 'dropout' layer sets a random selection of weights to zero during each training step, which helps prevent overfitting (*Srivastava et al., 2014*). Our implementation allows users to adjust the shape of the network, but current default settings use 10 dense layers of 256 nodes each with 'ELU' activations (*Clevert et al., 2015*) and a 25% dropout after the fifth layer. We describe performance under varying network width and depth in *Figure 5—figure supplement 1*. In general, we found that all networks with over four layers perform similarly.

We use the Adam optimizer (*Kingma and Ba, 2014*) with Euclidean distance as a loss function:

$$\text{loss} = \sqrt{(x_{\text{predicted}} - x_{\text{true}})^2 + (y_{\text{predicted}} - y_{\text{true}})^2}. \tag{1}$$

## Uncertainty and genome-wide variation

Individuals are born at a single location, but have inherited their genomes as a mosaic from ancestors spreading geographically into the past (as discussed in, for instance, *Wright, 1943*; *Kelleher et al., 2016*; *Bradburd and Ralph, 2019*). Any signal our method hopes to extract from the data must be due to geographic signal of recent ancestors shared between the test and training datasets. This suggests that any analogous method must quantify, roughly, 'which modern day populations are most similar to this genome?". The spatial spread of genetic relatedness both back in time from an individual's to its ancestors' locations and forward in time from ancestors to the present-day location of training samples means that even a perfect inference algorithm should have significant uncertainty associated with any predicted location from genetic data, and the magnitude of uncertainty should be in part a function of the dispersal rate of the population. In particular, no such method can infer locations more accurately than the mean dispersal distance, because in most cases an individual's genome is not informative about where they live relative to their parents. Besides this

fundamental limit to uncertainty, error in georeferencing of training samples and in model fitting will introduce additional prediction uncertainty.

We use a windowed analysis across the genome to describe this uncertainty, which is possible thanks to `Locator`'s computational efficiency. Genealogical relatedness on each contiguous stretch of genome can be described by a sequence of genealogical trees, separated by ancestral recombination events. By running Locator on a particular window of the genome, we restrict inference to a subset of these marginal trees, and hence to a subset of the genetic relationships between test and training samples. Predictions from different regions of the genome can then be visualized as a cloud of points, and the distribution of these points in space gives us a rough idea of the uncertainty associated with an individual-level prediction. Because windowed analyses involve repeated training runs from randomized starting parameters, they also help us to integrate over uncertainty associated with the model fitting process.

Some datasets lack the size or reference alignments necessary to conduct windowed analyses. In this case, we recommend uncertainty be assessed by training replicate models on bootstrapped samples drawn from a single set of unlinked SNPs (that is, resampling SNPs with replacement). Although this procedure does not reduce the number of marginal trees represented in the data, it does allow us to assess uncertainty associated with model training and prediction. In both cases, we summarize uncertainty in predicted locations by estimating a two-dimensional kernel density surface over a set of predicted locations, and provide plotting scripts to visualize the 95%, 50%, and 10% quantiles in geographic space (see *Figures 5–7* for examples). The location of an individual can then be predicted as either the location with highest kernel density (the modal prediction) or the geographic center of the cloud of predictions (the mean prediction).

We tested this approach in simulated data and in all empirical datasets. To explore factors affecting the accuracy of predicted locations generated from different regions of the genome, we also examined the relationship between recombination rate and test error from windowed *Locator* runs on human data from the HGDP panel (*Bergström et al., 2020*). Recombination rates for each window were estimated by averaging per-base rates from the HapMap project (*International HapMap Consortium, 2003*).

## Simulations

We first evaluated our method on genotypes from populations simulated by SLiM v3 (*Haller and Messer, 2019*), using the model of continuous space described in *Battey, 2019*. We simulated a $50 \times 50$ unit square landscape with expected density ($d$) of 5 individuals per unit area, resulting in census sizes of around 12,500. We varied the mean parent-offspring dispersal distance $\sigma$ across simulations from 0.45 to 3, to create populations with varying levels of isolation by distance. In terms of Wright's 'neighborhood size' (*Wright, 1946*), defined as $N_{loc} = 4\pi\sigma^2 d$, this yields populations with neighborhood sizes from 13 to 565. Each diploid individual carried two copies of a $10^8$ bp chromosome on which mutations and recombinations occured at a rate of $10^{-8}$ per bp per generation. Simulations were run until all extant individuals shared a single common ancestor within the simulation at all locations on the genome (i.e., the tree sequence had coalesced). 500 individuals were randomly sampled from the final generation of each simulation for use in model fitting.

We selected 50 individuals from each simulation as a validation set and ran Locator while varying the number of training samples from 10 to 450 and the number of SNPs from 100 to 100,000. The SNPs used were a subset sampled from the full genotype matrix without replacement and thus mimic the semi-random distribution of genome-wide SNPs generated by reduced-representation sequencing approaches like RADseq (*Etter et al., 2012*). To compare performance with an existing model-based approach, we also ran SPASIBA (*Guillot et al., 2016*) on the simulation with $\sigma = 0.44$ using 450 training samples and varying the number of SNPs from 100 to 100,000. Locator was run on a CUDA-enabled GPU and SPASIBA was run on 80 CPU cores. Last, we ran a windowed analysis on the $\sigma = 0.63$ (neighborhood size $\approx 25$) simulation in Locator using a 2Mbp window size (each window then contains $\approx$ 8000 SNPs).

## Empirical data

We applied Locator to three whole-genome resequencing datasets of geographically widespread samples: (1) 765 mosquitoes from the *Anopheles gambiae/coluzzii* species complex collected across

sub-Saharan Africa (*Anopheles gambiae 1000 Genomes Consortium et al., 2017*), (2) 5965 samples of the malaria parasite *Plasmodium falciparum* sequenced from human blood samples collected across Papua New Guinea, southeast Asia, sub-Saharan Africa, and northern South America (*Pearson et al., 2019*) and (3) whole-genome data for 56 human populations from the Human Genome Diversity Project (*Bergström et al., 2020*). Genotype calls for the *Anopheles* dataset are available at https://www.malariagen.net/data/ag1000g-phase1-ar3, for *P. falciparum* at https://www.malariagen.net/resource/26, and for human data at ftp://ngs.sanger.ac.uk/production/hgdp. We used VCF files as provided with no further postprocessing.

The *Plasmodium falciparum* dataset is unusual relative to our other empirical examples in that sequencing libraries were prepared from blood samples without filtering for coinfections or isolating individual *Plasmodium*. Sequence reads returned from short read sequencing then reflect the population of *Plasmodium* present in a human blood sample, or even multiple lineages of parasite if an individual is co-infected with multiple strains (*Zhu et al., 2019*), rather than individual *Plasmodium*. The VCFs we analyzed were prepared by aligning illumina short read sequences to the *Plasmodium falciparum* reference genome prepared by the Pf3K project (*Pf3K Consortium, 2016*; https://www.malariagen.net/data/pf3K-5), then calling SNPs in GATK (*McKenna et al., 2010*). Variant calls then represent the pool of mutations present in the infecting *Plasmodium* population rather than SNPs in a single individual. We used only field-collected samples from the 'analysis' set, as described in *Pearson et al., 2019*.

For the *Anopheles* dataset, we ran Locator in 2Mbp windows across the genome with a randomly selected 10% of individuals held out as a test set. We also ran `SPASIBA` on subsets sampled from the first five million base pairs of chromosome 2L while varying the number of SNPs from 100 to 100,000. For the *P. falciparum* dataset, we used 500 kb windows and held out 5% of samples from each collection locality as a test set. Last, for humans we used 10Mbp windows and selected three individuals from each HGDP population to hold out as a test set. Window sizes in each case were chosen to include roughly 100,000–200,000 SNPs per window. All empirical analyses were run with default settings (10×256 network size, patience 100, 25% dropout, a random 10% of training samples used for validation).

We also tested `Locator`'s performance with empirical data when the true location is not represented in the training sample. To do this, we ran a series of models on 10,000 SNPs randomly selected from the first 5Mbp of chromosome 2L in the *Anopheles* data. For each run, we held out all samples from a given sampling locality from the training set, then predicted the locations of these individuals using the trained model. We also tested this approach while holding out all samples collected in a given country, which eliminates even nearby localities from the training set.

## Data and code

Locator is implemented as a command-line program written in Python: www.github.com/kern-lab/locator. SNP calls for the *Anopheles* dataset are available at https://www.malariagen.net/data/ag1000g-phase1-ar3, for *P. falciparum* at https://www.malariagen.net/resource/26, and for the HGDP at ftp://ngs.sanger.ac.uk/production/hgdp. Code to run continuous-space simulations can be found at https://github.com/kern-lab/spaceness/blob/master/slim_recipes/spaceness.slim (*Battey, 2019*). This publication uses data from the MalariaGEN *Plasmodium falciparum* Community Project as described in *Pearson et al., 2019*. Statistical analyses and many plots were produced in R (*R Development Core Team, 2018*).

## Acknowledgements

We thank members of the Kern-Ralph co-lab, Daniel Schrider, Matthew Hahn, and Ethan Linck for comments and suggestions on this work, and Mara Lawniczak for the suggestion to look at the *Plasmodium* dataset. Comments from reviewers and editors at eLife significantly improved the final version of this study. CJB and ADK were funded by NIH award R01GM117241. PLR was funded in part by an I3 award from the University of Oregon.

## Additional information

### Funding

| Funder | Grant reference number | Author |
|---|---|---|
| National Institutes of Health | R01GM117241 | CJ Battey<br>Andrew D Kern |
| University of Oregon | I3 award | Peter L Ralph |

The funders had no role in study design, data collection and interpretation, or the decision to submit the work for publication.

### Author contributions

CJ Battey, Conceptualization, Resources, Data curation, Software, Formal analysis, Validation, Investigation, Visualization, Methodology, Writing - original draft, Writing - review and editing; Peter L Ralph, Conceptualization, Formal analysis, Supervision, Investigation, Methodology, Writing - review and editing; Andrew D Kern, Conceptualization, Software, Supervision, Funding acquisition, Investigation, Methodology, Project administration, Writing - review and editing

### Author ORCIDs

CJ Battey (iD) https://orcid.org/0000-0002-9958-4282
Andrew D Kern (iD) http://orcid.org/0000-0003-4381-4680

### Decision letter and Author response

Decision letter https://doi.org/10.7554/eLife.54507.sa1
Author response https://doi.org/10.7554/eLife.54507.sa2

## Additional files

### Supplementary files

• Supplementary file 1. Validation error in terms of map units and generations of mean population dispersal for Locator runs in simulations with 450 training samples and 100,000 SNPs. Note that while absolute error increases along with dispersal rate, it is roughly constant when expressed in terms of generations of mean dispersal.

• Supplementary file 2. Mean and median prediction error for Locator and SPASIBA run on simulations and Anopheles data as shown in *Figure 3*. Error is in terms of map units for simulated data (total landscape width = 50).

• Supplementary file 3. Test error for windowed analyses of empirical datasets using the location with highest kernel density and the centroid of per-window predictions, as *median (90% interval)*.

• Transparent reporting form

### Data availability

Locator is implemented as a command-line program written in Python: www.github.com/kern-lab/locator. SNP calls for the Anopheles dataset are available at https://www.malariagen.net/data/ag1000g-phase1-ar3, for *P. falciparum* at https://www.malariagen.net/resource/26, and for the HGDP at ftp://ngs.sanger.ac.uk/production/hgdp. Code to run continuous-space simulations can be found at https://github.com/kern-lab/spaceness/blob/master/slim_recipes/spaceness.slim. This publication uses data from the MalariaGEN Plasmodium falciparum Community Project as described in Pearson et al. (2019). Statistical analyses and many plots were produced in R (R Core Team, 2018).

The following previously published datasets were used:

| Author(s) | Year | Dataset title | Dataset URL | Database and Identifier |
|---|---|---|---|---|
| The Anopheles | 2015 | Ag1000G phase 1 AR3 data release | https://www.malariagen. | MalariaGEN, ag1000 |

| | | | net/data/ag1000g-phase1-ar3 | g-phase1-AR3 |
|---|---|---|---|---|
| gambiae 1000 Genomes Consortium | | | | |
| Plasmodium falciparum community project | 2019 | Plasmodium falciparum community project version 6 data release | https://www.malariagen.net/resource/26 | MalariaGEN, 26 |
| Bergström A, McCarthy SA, Hui R, Almarri MA, Ayub Q, Danecek P, Chen Y, Felkel S, Hallast P, Kamm J, Blanché H, Deleuze JF, Cann H, Mallick S, Reich D, Sandhu DA, Skoglund P, Scally A, Xue Y, Durbin R, Smith CT | 2019 | Insights into human genetic variation and population history from 929 diverse genomes | ftp://ngs.sanger.ac.uk/production/hgdp | HGDP, hgdp |

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

## Appendix 1

### Theoretical limits on accuracy

Suppose that we know the spatial locations of some relatives of a given individual, and want to predict the location of that focal individual. This is a best-case scenario for our actual problem, as in fact we would have to infer the degrees of relatedness of the reference set to the focal individual, but the calculations are useful in establishing a lower bound on the resolution of inference.

Suppose furthermore that the displacement in spatial position along each parent-child relationship has mean zero and variance $\sigma^2$, so that the net distance traveled along any path along $k$ links in the pedigree has mean zero and variance $k\sigma^2$. Given the location of $n$ relatives of a focal individual, a simple estimator of that individual's spatial location is simply the average of their locations. How well does this do?

We can associate each link between parent $p$ and child $c$ in the pedigree with the displacement between them, $X_{pc} = -X_{cp}$; we have assumed that $[X_{cp}] = \sigma^2$ for each. Suppose that the $i^{\text{th}}$ relative can be reached by traversing relatives $r_{i1}, \ldots, r_{ik_i}$, and so their location relative to the focal individual is $Y_i = X_{r_{i1}, r_{i2}} + \cdots + X_{r_{i(k_i-1)}, r_{ik_i}}$. To compute the variance of our estimator, $\bar{Y} = \sum_{i=1}^{n} Y_i/n$, let $n_{cp}$ be the number of $i$ for which $X_{cp}$ appears in the sum for $Y_i$, so that $\bar{Y} = \sum_{cp} n_{cp} X_{cp}/n$. Then, simply, $[\bar{Y}] = \sum_{cp} (n_{cp}/n)^2 X_{cp}$. For instance, if those relatives are all $2^k$ ancestors $k$ generations ago (i.e., the great $^{k-2}$-grandparents) of the focal individual, then each of the $2^\ell$ links between the $\ell^{\text{th}}$ and $(\ell-1)^{\text{th}}$ generations are traversed by $2^{k-\ell}$ of the paths, and so

$$\text{var}[\bar{Y}] = \sum_{\ell=1}^{k} 2^\ell \left(\frac{2^{k-\ell}}{2^k}\right)^2 \sigma^2 = (1 - 2^{-k})\sigma^2.$$

Clearly, with less full pedigree coverage and more distant relatives, the error would become worse, but it does not depend strongly on the degree of relatedness used: in general, using a few close or many distant relatives should give an estimate of location within some moderate factor of $\sigma$.

