## [Decision Letter]

**Acceptance summary:**

Your paper introduces a novel efficient tool for "placing samples on a map" using genetic variation – a fundamental problem of considerable practical importance.

**Decision letter after peer review:**

Thank you for submitting your article "Predicting Geographic Location from Genetic Variation with Deep Neural Networks" for consideration by *eLife*. Your article has been reviewed by Patricia Wittkopp as the Senior Editor, a Reviewing Editor, and three reviewers. The following individuals involved in review of your submission have agreed to reveal their identity: Christian Huber (Reviewer #1); Arbel Harpak (Reviewer #2); Leo Speidel (Reviewer #3).

The reviewers have discussed the reviews with one another and the Reviewing Editor has drafted this decision to help you prepare a revised submission.

Summary:

This paper describes an elegant and highly efficient method for assigning samples to geographic location based on polymorphism data.

Essential revisions:

The reviewers were unanimous in liking this paper, but were equally unanimous in thinking it needed more discussion/clarification of what the advantages and disadvantages of this machine learning approach is compared to explicitly model-based approaches. How should we interpret the results, what are suitable applications (beyond examples given), etc.

As the reviews are consistent and self-explanatory, they are attached in full, rather than condensed here. This also goes for minor comments.

Reviewer #1:

The authors present a new, neural network (NN) based approach for predicting the geographic location of a sample based on its genetic information and a reference dataset of multiple samples with known genetic and geographic information. The new method, Locator, is compared to a probabilistic approach that explicitly models the geographic distribution of alleles.

One major advantage of the new approach is the dramatic increase in speed, particularly when dealing with very large datasets. The simulations and empirical analyses make a lot of sense. I further appreciate the theoretical and simulation-based insight that dispersal distance is the single limiting factor for the accuracy of the inference, assuming datasets are moderately large. I strongly support publication of the study in *eLife*. However, I think the manuscript could improve by providing a better presentation of advantages and disadvantages of this type of machine learning approach compared to model-based (i.e. statistical) approaches. I suspect that model-based approaches such as SPASIBA should work better for out-of-sample prediction (e.g. for geographic regions that are not covered by the training sample) and have a clearer approach to quantifying uncertainty (i.e. confidence intervals), at least when model assumptions are correct. Specific comments and examples are below.

1) I wonder about cases where the distribution of the species is continuous across space, but the sampling of the training sample is strongly clustered, which seems highly relevant for many empirical cases where samples are rarely sampled uniformly across species range (e.g., the Anopheles data). Further, it seems that the clustering of the training samples affects the prediction of Locator almost like a strong "prior", i.e. as if it is implicitly assumed that the new sample most likely also comes from one of these clusters. This might explain why Locator is superior to SPASIBA for the Anopheles data but not for the simulations (Figure 3) since SPASIBA does not have a prior on location but assumes that the new sample could come from any arbitrary location. However, this might also imply that SPASIBA is better for out-of-sample prediction, i.e. when the spatial location of the investigated sample is different from the locations of the clustered training data. Related to this, in the discussion the authors state that they found that "the greatest increase in performance relative to the model-based approach is in empirical data for which the assumption of smooth variation in allele frequencies across the landscape is unlikely to hold, such as the complex multi-species Anopheles sample analyzed here (Figure 3)". My alternative interpretation here is that this has nothing to do with smooth variation in allele frequency, but is a result of the strong clustering of the training sample, and the fact that the test samples are also sampled from these cluster locations. In fact, one of the results (subsection "Effects of Unsampled Populations") is that prediction error is much higher for individuals from regions excluded from training, and that "predicted locations appear to project towards the nearest locality included in the training set", which is supporting my interpretation. I thus would change the discussion regarding the mentioned statement and bring up the problem of clustered sampling locations of the training sample. It might be out of the scope of the current study, but testing both SPASIBA and Locator on spatial simulations with clustered training samples but uniformly sampled test samples would also be informative regarding this question.

2) The statistical method SPASIBA allows to compute a likelihood surface of the location of the tested individual, i.e. it provides a statistical way of quantifying uncertainty in the estimate. The Locator method does not allow such a quantification. However, the authors suggest to run Locator on multiple 2 Mb windows across the genome and use the predicted locations from such multiple runs to quantify the uncertainty, for example by plotting contours of the 95% quantiles of a two-dimensional kernel density of these locations. This approach certainly gives some idea about the spatial extent of uncertainty (i.e. if there are certain spatial directions where estimating the location is harder than for other directions). However, it seems important to point out that these contours are not proper confidence or credibility intervals to avoid any confusion. There is no guarantee that the 95% contours capture the true location 95% of times. Further, the contours might look very different depending on the choice of the window size. The authors should provide more explicit guidelines about how to (or not to) interpret these contours, and also guide the choice of window size (e.g., why 2 Mb? Is there an optimal window size?).

Reviewer #2:

Battey et al., develop a software to predict geographic location from genotype data. They use the fact that their method is fast to quantify individual prediction uncertainty and describe "mosaic ancestry" by predicting locations separately in windows across the genome.

The manuscript and code are overall clear and the work is well-rounded, with applications to simulations and three real data sets and a thoughtful discussion of the method's performance on each. The fact that LOCATOR works directly on a VCF and a coordinates file, is very fast, and quantifies individual level uncertainty-should all be very appealing to users.

I have some doubts are about the utility of the method and about the neural net approach. What are some possible motivations / applications beyond the trafficked elephant ivory example? Assuming that there are some important applications, is the neural net approach really preferable to a generative model? Throughout, the authors articulate interesting insight (e.g. the reasons underlying variance in estimates across genomic windows; the way dispersal distance sets a lower bound on accuracy) that they cannot fully quantify with their method due to its black-box nature. Relatedly, the authors demonstrate that the performance is superior to SPASIBA when the input is raw, unphased genotype data. While this may represent the current state of the art, it seems to me plausible that in the future we will separate the learning of genealogical trees from learning dispersal patterns. I wonder, for example, if feeding the network with inferred tree sequences (e.g. tsInfer) would lead to much better performance, as it disentangles these two learning steps. To be clear, I am not asking the authors to develop a new method, but only to modify their discussion to either explain the benefits of their steering away from the generative model; or alternatively say that down the line, a generative approach is more fruitful.

Reviewer #3:

The manuscript describes a method for predicting geographic location using a simple, fully connected deep neural network which is trained on labelled samples of known geographic location. I am particularly impressed that the method can be applied to genomic subregions, quantifying uncertainty about predictions as well as variation of ancestry along the genome due to recent admixture or other evolutionary processes influencing genetic structure.

The presented method is a significant advance from existing methods, which are substantially slower and impose stronger modelling assumptions. Accuracy appears comparable to existing methods on simulated data (with one exception, see below), while on real data the presented method outperforms existing approaches.

The manuscript is very well written; details of the method were clear. While I did not attempt to do so, I believe that analyses were explained in sufficient detail for reproducing the results.

I believe that the work is of interest to the readership of *eLife* and I would recommend it for publication. Below, I am listing a few comments I have after reading the manuscript.

- Prediction of geographic location has many similarities with (recent) ancestry inference, and can be seen as a projection of genetic structure onto a geographic landscape (where genetic structure may be affected by many processes). It would be great if the authors could include a discussion putting this work into context with previous work on genetic structure, in particular methods that also don't require discrete populations, e.g., PCA on various types of matrices (genotype, chromopainter coancestry matrices etc) (genes mirror geography!). I do think and like that projecting structure on geography can improve interpretability compared to these methods, however perhaps adds constraints on the projection space that other methods may not have/or differ from other methods?

- I believe that a potential weak point of the current manuscript is that it is unclear how to interpret these predictions (apart from the naive interpretation of geographic origin of ancestors). In particular, I believe that the impact of this paper would be very much elevated if the authors could include examples where geographic location can be used for inference of some underlying biological process, e.g., selection acting on the genome or regions with reduced genetic diversity. Is there a difference in predicted geographic location in coding/non-coding regions (such as coding regions have lower variance of prediction for instance)?

- Is it possible to estimate dispersal rates from these predictions and their uncertainty? (which may be the obvious parameter one might want to learn about an organism/group of interest).

- I like that the authors provided an analysis where some samples were dropped from the training step, to illustrate the effect of unsampled groups. In practise, is there a way to learn from the data whether the reference/training samples are representative for the sample I want to do prediction for? E.g., one could think about having multiple neural networks that are trained on subsets of samples and compare predictions.

- What is the reason for windowing according to number of base-pairs instead of recombination rate (cM), especially given that Figure 8 shows a higher prediction error in regions of lower recombination rate (and fewer genealogical trees).

Overall, I very much enjoyed reading this paper. Out of interest and with no need to mention or implementing this in the current paper – If the underlying genealogies were known, would you be able to predict geographic locations of most recent common ancestors? You could then extract information about directional migration from this for instance I believe?

---

## [Author Response]

Essential revisions:The reviewers were unanimous in liking this paper, but were equally unanimous in thinking it needed more discussion/clarification of what the advantages and disadvantages of this machine learning approach is compared to explicitly model-based approaches. How should we interpret the results, what are suitable applications (beyond examples given), etc.As the reviews are consistent and self-explanatory, they are attached in full, rather than condensed here. This also goes for minor comments.

Thanks to the editors and reviewers for your thorough comments. In this revision we have added two new analyses looking at (1) fixed- vs recombination-distance-based windows, and (2) the EDAR locus as an example of a well-studied genomic region with a steep geographic cline. We also added to the end of our Discussion section to explore the general utility of machine learning approaches in population genetics. We also made minor figure edits and clarified several methodological points and syntax issues. Specific responses for each reviewer comment are printed below.

Reviewer #1:The authors present a new, neural network (NN) based approach for predicting the geographic location of a sample based on its genetic information and a reference dataset of multiple samples with known genetic and geographic information. The new method, Locator, is compared to a probabilistic approach that explicitly models the geographic distribution of alleles.One major advantage of the new approach is the dramatic increase in speed, particularly when dealing with very large datasets. The simulations and empirical analyses make a lot of sense. I further appreciate the theoretical and simulation-based insight that dispersal distance is the single limiting factor for the accuracy of the inference, assuming datasets are moderately large. I strongly support publication of the study in eLife. However, I think the manuscript could improve by providing a better presentation of advantages and disadvantages of this type of machine learning approach compared to model-based (i.e. statistical) approaches. I suspect that model-based approaches such as SPASIBA should work better for out-of-sample prediction (e.g. for geographic regions that are not covered by the training sample) and have a clearer approach to quantifying uncertainty (i.e. confidence intervals), at least when model assumptions are correct. Specific comments and examples are below.

Thank you for your comments. We have expanded the general discussion of the utility of machine learning approaches in population genetics at the end of the Discussion section, and respond to specific comments below.

1) I wonder about cases where the distribution of the species is continuous across space, but the sampling of the training sample is strongly clustered, which seems highly relevant for many empirical cases where samples are rarely sampled uniformly across species range (e.g., the Anopheles data). Further, it seems that the clustering of the training samples affects the prediction of Locator almost like a strong "prior", i.e. as if it is implicitly assumed that the new sample most likely also comes from one of these clusters. This might explain why Locator is superior to SPASIBA for the Anopheles data but not for the simulations (Figure 3) since SPASIBA does not have a prior on location but assumes that the new sample could come from any arbitrary location. However, this might also imply that SPASIBA is better for out-of-sample prediction, i.e. when the spatial location of the investigated sample is different from the locations of the clustered training data. Related to this, in the discussion the authors state that they found that "the greatest increase in performance relative to the model-based approach is in empirical data for which the assumption of smooth variation in allele frequencies across the landscape is unlikely to hold, such as the complex multi-species Anopheles sample analyzed here (Figure 3)". My alternative interpretation here is that this has nothing to do with smooth variation in allele frequency, but is a result of the strong clustering of the training sample, and the fact that the test samples are also sampled from these cluster locations. In fact, one of the results (subsection "Effects of Unsampled Populations") is that prediction error is much higher for individuals from regions excluded from training, and that "predicted locations appear to project towards the nearest locality included in the training set", which is supporting my interpretation. I thus would change the discussion regarding the mentioned statement and bring up the problem of clustered sampling locations of the training sample. It might be out of the scope of the current study, but testing both SPASIBA and Locator on spatial simulations with clustered training samples but uniformly sampled test samples would also be informative regarding this question.

This is a good point, and the analogy to a strong prior given spatially concentrated sampling seems right to us. We have edited the second paragraph of the Discussion section to include this point.

2) The statistical method SPASIBA allows to compute a likelihood surface of the location of the tested individual, i.e. it provides a statistical way of quantifying uncertainty in the estimate. The Locator method does not allow such a quantification. However, the authors suggest to run Locator on multiple 2 Mb windows across the genome and use the predicted locations from such multiple runs to quantify the uncertainty, for example by plotting contours of the 95% quantiles of a two-dimensional kernel density of these locations. This approach certainly gives some idea about the spatial extent of uncertainty (i.e. if there are certain spatial directions where estimating the location is harder than for other directions). However, it seems important to point out that these contours are not proper confidence or credibility intervals to avoid any confusion. There is no guarantee that the 95% contours capture the true location 95% of times. Further, the contours might look very different depending on the choice of the window size. The authors should provide more explicit guidelines about how to (or not to) interpret these contours, and also guide the choice of window size (e.g., why 2 Mb? Is there an optimal window size?).

We added a note to the relevant Results section explaining differences between window prediction intervals and confidence intervals (ie we think they’re best interpreted as “x% of the genome likely originated in this interval”).

Reviewer #2:Battey et al., develop a software to predict geographic location from genotype data. They use the fact that their method is fast to quantify individual prediction uncertainty and describe "mosaic ancestry" by predicting locations separately in windows across the genome.The manuscript and code are overall clear and the work is well-rounded, with applications to simulations and three real data sets and a thoughtful discussion of the method's performance on each. The fact that LOCATOR works directly on a VCF and a coordinates file, is very fast, and quantifies individual level uncertainty-should all be very appealing to users.I have some doubts are about the utility of the method and about the neural net approach. What are some possible motivations / applications beyond the trafficked elephant ivory example? Assuming that there are some important applications, is the neural net approach really preferable to a generative model? Throughout, the authors articulate interesting insight (e.g. the reasons underlying variance in estimates across genomic windows; the way dispersal distance sets a lower bound on accuracy) that they cannot fully quantify with their method due to its black-box nature. Relatedly, the authors demonstrate that the performance is superior to SPASIBA when the input is raw, unphased genotype data. While this may represent the current state of the art, it seems to me plausible that in the future we will separate the learning of genealogical trees from learning dispersal patterns. I wonder, for example, if feeding the network with inferred tree sequences (e.g. tsInfer) would lead to much better performance, as it disentangles these two learning steps. To be clear, I am not asking the authors to develop a new method, but only to modify their discussion to either explain the benefits of their steering away from the generative model; or alternatively say that down the line, a generative approach is more fruitful.

Thank you for your comments. We have expanded our Discussion section to talk more generally about the utility and desirability of machine-learning vs statistical approaches – see the new final two paragraphs.

Reviewer #3:The manuscript describes a method for predicting geographic location using a simple, fully connected deep neural network which is trained on labelled samples of known geographic location. I am particularly impressed that the method can be applied to genomic subregions, quantifying uncertainty about predictions as well as variation of ancestry along the genome due to recent admixture or other evolutionary processes influencing genetic structure.The presented method is a significant advance from existing methods, which are substantially slower and impose stronger modelling assumptions. Accuracy appears comparable to existing methods on simulated data (with one exception, see below), while on real data the presented method outperforms existing approaches.The manuscript is very well written; details of the method were clear. While I did not attempt to do so, I believe that analyses were explained in sufficient detail for reproducing the results.I believe that the work is of interest to the readership of eLife and I would recommend it for publication. Below, I am listing a few comments I have after reading the manuscript.

Thank you for your comments. In this revision we have expanded our discussion to talk more generally about the utility of process-agnostic models in population genetic inference, and made a number of small edits to syntax and figures. Specific responses are below.

- Prediction of geographic location has many similarities with (recent) ancestry inference, and can be seen as a projection of genetic structure onto a geographic landscape (where genetic structure may be affected by many processes). It would be great if the authors could include a discussion putting this work into context with previous work on genetic structure, in particular methods that also don't require discrete populations, e.g., PCA on various types of matrices (genotype, chromopainter coancestry matrices etc) (genes mirror geography!). I do think and like that projecting structure on geography can improve interpretability compared to these methods, however perhaps adds constraints on the projection space that other methods may not have/or differ from other methods?

We like this suggestion. To address this we have added the following text to the Discussion section: “The correspondence between our explicitly geographic method and unsupervised clustering or dimensionality reduction methods highlights the implicit prior assumption of genetic-geographic correlation often made when interpreting the output of STRUCTURE or PCA. Rather than mapping population structure to geography as a post-hoc interpretation, Locator and other continuous assignment methods directly incorporate space in the model.”

- I believe that a potential weak point of the current manuscript is that it is unclear how to interpret these predictions (apart from the naive interpretation of geographic origin of ancestors). In particular, I believe that the impact of this paper would be very much elevated if the authors could include examples where geographic location can be used for inference of some underlying biological process, e.g., selection acting on the genome or regions with reduced genetic diversity. Is there a difference in predicted geographic location in coding/non-coding regions (such as coding regions have lower variance of prediction for instance)?

Thanks for this suggestion. We have now expanded on our analysis of association or error with recombination rate to look at different windowing designs, and pulled out a case study of the EDAR locus in Eurasia as an example of a highly geographically differentiated region of the genome that readers familiar with the human genetics literature in particular may be familiar with.

- Is it possible to estimate dispersal rates from these predictions and their uncertainty? (which may be the obvious parameter one might want to learn about an organism/group of interest).

In simulations we found that error was roughly constant when expressed in units of population mean dispersal distance (3 – 4 generations of dispersal), so it may be possible to work backwards from observed validation error in empirical data to estimate a dispersal kernal for the organism. However, we didn’t design the method with this use in mind and haven’t tested extensively enough to be confident in using it that way, so we decided to leave this out of the main text.

- I like that the authors provided an analysis where some samples were dropped from the training step, to illustrate the effect of unsampled groups. In practise, is there a way to learn from the data whether the reference/training samples are representative for the sample I want to do prediction for? E.g., one could think about having multiple neural networks that are trained on subsets of samples and compare predictions.

The multiple networks idea is an interesting one that we hadn’t considered. The closest approach that we’ve worked on here is in small samples (i.e. *<*40 training samples), where randomly splitting training/validation samples can result in some regions under- or overrepresented in the training set. On the Locator documentation (i.e., the github readme) we now included a section suggesting users fit models in a loop with different seeds to get an ensemble of models trained on different subsets of their reference data. This doesn’t quite get at whether or not each training set is representative, but does help average over some of the noise from the training / validation split. In this revision we added a paragraph to the Materials and methods section suggesting this approach to users with small sample sizes.

- What is the reason for windowing according to number of base-pairs instead of recombination rate (cM), especially given that Figure 8 shows a higher prediction error in regions of lower recombination rate (and fewer genealogical trees).

Good point. We have now added a section comparing fixed versus per-centimorgan windowing, and looking a little deeper at genomic factors that could affect prediction accuracy. Perhaps unsurprisingly the error of centroid estimates is extremely similar using bp or cm windows, but the distribution along the genome is quite different in a way that seems to reflect the association with recombination rate. Overall, the variance in per-window prediction error is lower for recombination-distance based windows. This mostly seems to reflect centromeres, whose presence predicts significant proportions of variance in error using fixed windows but not in centimorgan windows.

Overall, I very much enjoyed reading this paper. Out of interest and with no need to mention or implementing this in the current paper – If the underlying genealogies were known, would you be able to predict geographic locations of most recent common ancestors? You could then extract information about directional migration from this for instance I believe?

Interesting idea – I think the answer is yes if you make some assumptions about the dispersal kernal (the model we sketch out in the appendix would get you part of the way there). We are also aware of another group with a method in the works that does something like this and are excited to see it!